


**Detection of Stratospheric Air Intrusion Events From Ground-based**
**High-resolution $^{10}$Be/$^{7}$Be by Accelerator Mass Spectrometry**
Xu-Ke Liu[1,2,3,5,#], Yun-Chong Fu[1,2,4,#,*], Li Zhang[1,2], George S. Burr[1], Yan-Ting Bi[5], Guo-Qing
Zhao[1,2]
*1. State Key Laboratory of Loess and Quaternary Geology, Institute of Earth Environment,*
*Chinese Academy of Sciences (IEECAS), Xi'an 710061, China*
*2. Shaanxi Key Laboratory of Accelerator Mass Spectrometry Technology and Application, Xi'an*
*AMS Center of IEECAS, Xi'an 710061, China*
*3. University of Chinese Academy of Sciences, Beijing 100049, China*
*4. Institute of Global Environmental Change, Xi'an Jiaotong University, Xi'an 710049, China*
*5. Xi'an Institute for Innovative Earth Environment Research, Xi'an, 710061, China*
*#The authors contribution equally to this work and should be considered co-first authors*
*Corresponding author: Yun-Chong Fu, E-mail: fuyc@ieecas.cn.*
**Abstract**
Locally rapid stratospheric air intrusions facilitate the transport of stratospheric
material to the troposphere. Long-term continuous monitoring of such events by
traditional techniques, such as sounding technology, is challenging. Beryllium-7 ($^{7}$Be)
and beryllium-10 ($^{10}$Be) offer an alternative. These isotopes are formed by cosmic
rays and are mainly produced in the lower stratosphere and upper troposphere. Due to
their similar geochemical properties and substantial difference in half-lives favor
relatively high $^{10}$Be/$^{7}$Be ratios in the stratosphere, as compared to the troposphere.
Monitoring surface $^{10}$Be/$^{7}$Be ratios affords a potential means to identify stratospheric
air intrusions. However, high temporal resolution $^{10}$Be/$^{7}$Be observational records must
be taken and corrected for dust-borne $^{10}$Be to identify stratospheric air intrusions. In


this study, we use Accelerator Mass Spectrometry to measure both $^{7}Be$ and $^{10}Be$ in
rain and aerosol (down to ~ 200 cubic meters air) with an error of ~ 1.5%. We correct
for dust-borne $^{10}Be$ using soil Al. This method provides precise measurements with
daily resolution. We present annual beryllium isotopes ($^{7}Be$, $^{10}Be$, and $^{10}Be/^{7}Be$ ratio)
record for the Chinese Loess Plateau that includes several regional sites. We show that
for the city of Xi'an, the proportion of dust-borne (resuspended) $^{10}Be$ was ~24% in
2020/21. Our results confirm that stratospheric air intrusion events in the Loess
Plateau are frequent and rapid throughout the year and are strongest in the spring
(March-July), when $^{10}Be/^{7}Be$ values were observed to increase about a factor of 3.
Even in winter, weaker stratospheric air intrusion events can be detected. Calculated
$\Delta(^{10}Be/^{7}Be)$ values in winter suggest stratospheric ozone transport can lead to an
~25% cumulative increase the surface ozone.

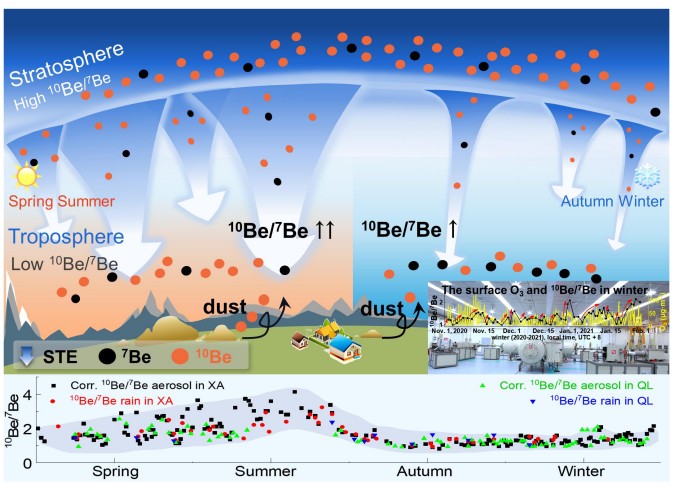

40                                    Abstract graph

**Key words:** Cosmogenic nuclides, Stratospheric air intrusion, $^{10}Be/^{7}Be$ ratio,





Accelerator mass spectrometry, Ozone transport, Loess Plateau

**1. Introduction**
Stratospheric air intrusions are a vital process in stratosphere-troposphere
exchange (STE), providing a transport mechanism for stratospheric materials to the
ground (Holton et al., 1995;Randel et al., 2010). Local stratospheric air intrusions are
difficult to quantify and may have hard-to-capture frequent intrusion events
(Hodnebrog et al., 2016;Sudo et al., 2003). Although these local stratospheric air
intrusions are relatively weak, they can still facilitate the transport of natural and
anthropogenic trace chemicals and significantly affects the atmospheric chemistry,
atmospheric heat budget, and radiation properties. For example, stratospheric air
intrusions can facilitate ozone ($O_3$) transport to the troposphere (Baylon et al., 2016).
Tropospheric $O_3$ affects human health and ecological environments. Due to the
scarcity of observational data in the lower stratosphere and upper troposphere, there
are few observational studies on this process. Traditional observation methods of
stratospheric air intrusion mainly rely on sounding technology with multi-element
sensors (Fischer et al., 2008). However, the observation process is complicated, and it
is challenging to observe continuously due to the influence of weather, let alone
accurate capture of frequent intrusion events in local areas.
Meteoric $^7Be$ and $^{10}Be$ are produced by cosmic rays interacting with oxygen and
nitrogen, mainly in the lower stratosphere and upper troposphere (Brown et al., 1989).
After their formation, these beryllium isotopes are rapidly absorbed and transported





by submicron aerosols (Lal et al., 1958), and it is generally believed that
ground-based $^7$Be analyses offer a means of identifying and tracing stratospheric air
intrusions (Bhandari et al., 1966;Hernandez-Ceballos et al., 2017). Using $^7$Be alone is
complicated by tropospheric weather conditions and the latitudinal dependence of $^7$Be
production at different latitudes (Masarik and Beer, 1999). Using $^{10}$Be/$^7$Be ratios can
avoid these effects, making it more accurate and sensitive than $^7$Be concentrations
alone to trace stratospheric air intrusions (Jordan et al., 2003). The transport and
deposition processes of $^7$Be and $^{10}$Be are same, and their production rates ratio at each
latitude is consistent (Masarik and Beer, 1999). The two isotopes have different
residence times in the stratosphere and troposphere and the very different half-lives:
$^7$Be ($T_{1/2}$ = 53.29 d) and $^{10}$Be ($T_{1/2}$ = 1.36 Ma). These factors make stratospheric
$^{10}$Be/$^7$Be ratios much larger than $^{10}$Be/$^7$Be ratios observed near the surface air (Jordan
et al., 2003). When stratospheric air intrudes, the $^{10}$Be/$^7$Be ratio in the atmosphere at
the earth's surface increases (Brown et al., 1989).

$^7$Be and $^{10}$Be are typically measured by gamma spectrometry and accelerator

mass spectrometry (AMS), respectively. Due to the low counting efficiency of the
gamma spectrometry, large samples are required (> $10^3$ m$^3$ air or > 10 L rain), as well
as lengthy measurements (> 1 day) to obtain optimal measurement uncertainties
(Tiessen et al., 2019;Yamagata et al., 2019). It is even more challenging to obtain $^7$Be
data with low uncertainty at a daily resolution based on gamma spectrometry,
especially in short-period field sampling and quickly stratospheric air intrusion event
research. Measuring $^7$Be by AMS significantly reduces sample size requirements and




improves the detection limit by approximately 10-100 times (Raisbeck and Yiou,
1988). Hence, we measure both isotopes with AMS.

Due to the high abundance of $^{10}$Be in topsoil, resuspended dust-borne $^{10}$Be must

be accounted (Czymzik et al., 2018;Yamagata et al., 2010). In particular, corrections
are necessary for STE studies in high dust areas such as the Chinese Loess Plateau
(one of the leading dust source areas in Asia (Zhang et al., 1997)), that reflect
$^{10}$Be/$^{7}$Be from dry deposition (Heikkila et al., 2013). We have devised a means of
making these corrections based in part on our previous studies of $^{10}$Be in loess (Beck
et al., 2018). In addition, high-precision $^{7}$Be and $^{10}$Be data can contribute to our
understanding of atmospheric material transport and circulation (Chham et al.,
2019;Liu et al., 2022), solar variability (Adolphi et al., 2014), and paleoclimate (Beck
et al., 2018).

In this work, we use ground-based $^{10}$Be/$^{7}$Be measurements as a means of

documenting stratospheric air intrusion events through time. In order to achieve this,
we first developed an AMS measurement method to quantify $^{7}$Be and $^{10}$Be together
(named $^{7}$Be-$^{10}$Be-AMS). Next, we identified an analytical means to remove the
resuspended dust-borne $^{10}$Be component, ultimately enabling the acquisition of
$^{10}$Be/$^{7}$Be records with low uncertainty and high temporal resolution in various
environments. We made numerous analyses of aerosol and rain samples ($n$ = 398)
from the Loess Plateau, and an annual $^{10}$Be/$^{7}$Be record with the daily resolution was
obtained for the first time. We show a relationship between these stratospheric air
intrusions in winter with surface $O_3$.



## 2. Methods

### 2.1 Sample collection

From May 2020 to June 2021, aerosol and rainwater samples from multiple locations were collected. These sampling points include (Fig. S1): Xi'an (XA, 34°13′20″N, 109°00′18″E) and Qinling (QL, 34°3′43″N, 108°20′48″E) for long-term monitoring. Zhongwei (ZW, 37°30′57″N, 105°11′8″E), Taiyuan (TY, 37°48′26″N, 112°34′55″E), Chengdu (CD, 30°56′44″N, 103°40′41″E), Nanning (NN, 22°50′18″N, 108°16′51″E), and Lianyungang (LYG, 34°46′4″N, 119°26′55″E) for short-term intermittent monitoring. Aerosols and rainwater were collected at XA and QL, while only aerosols were collected at the remaining sampling sites.

Total suspended particulate (TSP) samples passed through a large-flow particle collector (TH-1000CII, Wuhan Tianhong Instrument. Co., Ltd.) and a portable small-flow particle collector (TH-150H, Wuhan Tianhong Instrument. Co., Ltd.), prior to collection on polypropylene filter membranes. Each collection period was 24 h and sampled approximately 200-500 m³ air. For rain sample collection, the initial rainfall in the first half-hour was removed to avoid interference from suspended dust. Rainwater samples were collected and stored separately in acidified polyethylene containers to prevent beryllium in the rainwater from adsorbing on the container walls. Detailed rain sample collection information is described elsewhere (Zhang and Fu, 2017). Among multiple sampling points, the meteorological data collector (HOBO-U30 Station) established at the XA sampling site was used to accurately monitor meteorological data such as wind speed, precipitation, and solar radiation





intensity. Surface $O_3$ concentration data at the XA site was downloaded from the
government website https://www.aqistudy.cn/. The sampling times provided in this
article are based on local time (UTC+8).

**2.2 Extraction of Be and Al in the sample and preparation of the BeO target**

$^7$Be and $^{10}$Be targets were prepared according to established experimental
procedures for loess $^{10}$Be (Zhou et al., 2007), and rainwater $^7$Be and $^{10}$Be (Zhang and
Fu, 2017). The existing experimental process for aerosol samples followed 3 steps
(Fig. S2): 1) $^7$Be and $^{10}$Be extraction; 2) ion-exchange separation and purification; and
3) BeO preparation. The first step removes organic at 600 °C, and the remaining
aerosol component is dissolved in acid. Next, $Be(OH)_2$ is obtained by reaction with a
weak base ($NH_3 \cdot H_2O$), and the precipitate is oxidized to BeO at 900 °C. The Al
content of aerosols extracted by acid immersion was measured by ICP-AES
(ULTIMA-2, HORIBA Jobin Yvon, France) according to the method proposed by
Yamagata et al. (2010).

**2.3 AMS analysis of $^7$Be and $^{10}$Be**

$^7$Be and $^{10}$Be were analyzed in the same target by the 3 MV multi-nuclide AMS
at the Xi'an Accelerator Mass Spectrometry Center, Institute of Earth Environment,
Chinese Academy of Sciences. The analysis method follows the approach we
established in 2017 (Zhang and Fu, 2017), with some subsequent refinements. In
particular, the transmission efficiency after the second stripping foil was substantially
improved, reaching approximately 24%, which greatly improved the total
transmission efficiency and further improved the analysis precision. $^{10}Be^{4+}$ or $^7Be^{4+}$



were analyzed in a gas ionization detector (energy spectrum shown in Fig. S3).
Measurement details are given in the Methods section of the Supplementary
Information.

Data quality was assessed considering chemical preparation, measurement

uncertainties, and parallel sample monitoring results. A threshold sample recovery rate
for the chemical treatment process is maintained at > 80%. Each measurement batch
included a standard sample and a blank sample. Standard samples were used for
calibration, and blank samples were used for monitoring procedures. The results for
the blank samples were well below measured sample values (~$10^3$ times). The
measurement results of parallel samples are consistent within 1σ (Table S1).

AMS [7]Be measurements were cross-checked by gamma spectrometry. Large

samples (approximately 2000 m$^3$ air) were collected, and polypropylene membranes
with high aerosol concentrations were selected for comparison. About 1/4 of them
were analyzed by [7]Be-AMS. The [7]Be in the remaining 3/4 filter membranes were
measured using a high purity germanium (HpGe) detector. The results show that the
AMS results were consistent with the measurement HpGe detector results (Table S2).
Furthermore, for the same samples, the uncertainties for samples measured by AMS
were uniformly lower than those measured by HpGe detector (detailed measurement
information of the HpGe detector is included in the support information).
**2.4 Quantification of the resuspended dust contribution to [10]Be**

The resuspended dust-borne [10]Be contribution was corrected as:

$[^{10}Be]_{corr} = [^{10}Be]_{bulk} - [^{10}Be]_{dust}$ . The dust proportion was estimated from the





aluminum content $P = \dfrac{[Al]_{aerosol}}{Eff \cdot [Al]_{soil}}$ , and this value was used to calculate the dust
component from the soil [10]Be: $[^{10}Be]_{dust} = P \cdot [^{10}Be]_{soil}$ (Yamagata et al., 2010):
$$[^{10}Be]_{corr} = [^{10}Be]_{bulk} - \frac{[Al]_{aerosol}}{Eff \cdot [Al]_{soil}} \cdot [^{10}Be]_{soil} \quad (1)$$

where $[Al]_{aerosol}$ is the aerosol Al concentration extracted by acid dissolution

(g·m[-3]). Eff is the acid leaching efficiency of Al in aerosols, which is 51% (Yamagata
et al., 2010), as determined by comparisons between leached samples analyzed by
ICP-AES and a large number of samples analyzed by NAA. $[Al]_{soil}$ is the Al content
of the topsoil (wt%); $[^{10}Be]_{soil}$ is the [10]Be concentration of the topsoil (atoms·g[-1]).

For $[^{10}Be]_{soil}$ in equation (1), XA, QL, and TY belong to the Loess Plateau.

According to our previous results, this value is $2.13 \cdot 10^8$ atoms·g[-1] (Zhou et al., 2007).
According to survey results of [10]Be in topsoil (Yi et al., (2019b) (Fig. S1), the
$[^{10}Be]_{soil}$ values of ZW, LYG, CD, and NN are $11.70 \cdot 10^8$ atoms·g[-1], $3.75 \cdot 10^8$ atoms·g[-1],
$4.45 \cdot 10^8$ atoms·g[-1], and $2.50 \cdot 10^8$ atoms·g[-1], respectively. The $[Al]_{soil}$ contents are 7.41
wt% (Xiong et al., 2010), 7.57 wt% (Qiu et al., 2014), 8.32 wt% (Tan et al., 2013),
and 10.50 wt% (Qiu et al., 2014) on the Loess Plateau, LYG, CD, and NN,
respectively.

Researchers have shown that removing the initial precipitation (the first

half-hour) for rain samples reduces the resuspended dust-borne [10]Be (Graham et al.,
2003) and allows for a straightforward estimation of [10]Be from wet deposition.
However, in this approach, the particles associated with the STE source are discarded,
obscuring the relationship with aerosol [10]Be. Therefore, to verify our dust-borne [10]Be



corrections, we compare $^{10}$Be/$^{7}$Be of aerosols and precipitation.

**3. Results and discussion**


**3.1 Observations of atmospheric $^{7}$Be and $^{10}$Be deposited on the Chinese Loess**


**Plateau**


Measured atmospheric and rainwater $^{7}$Be and $^{10}$Be concentrations, and $^{10}$Be/$^{7}$Be
ratios from the XA and QL sites (May 2020 to June 2021) are presented in Fig. 1. At
the XA site, the average annual aerosol $^{7}$Be concentration was $(3.80 \pm 0.06) \cdot 10^{4}$
atoms·m$^{-3}$, and the average annual aerosol $^{10}$Be concentration was $(8.09 \pm 0.13) \cdot 10^{4}$
atoms·m$^{-3}$. XA rainwater $^{7}$Be values averaged $(4.00 \pm 0.16) \cdot 10^{4}$ atoms·g$^{-1}$, and $^{10}$Be
values averaged $(6.42 \pm 0.26) \cdot 10^{4}$ atoms·g$^{-1}$. At the QL site (Fig. 1b), the average
annual aerosol $^{7}$Be concentration was $(4.08 \pm 0.07) \cdot 10^{4}$ atoms·m$^{-3}$, and the average
annual aerosol $^{10}$Be concentration was $(6.76 \pm 0.11) \cdot 10^{4}$ atoms·m$^{-3}$. QL rainwater $^{7}$Be
values averaged $(4.86 \pm 0.19) \cdot 10^{4}$ atoms·g$^{-1}$, and $^{10}$Be values averaged $(6.35 \pm$
$0.24) \cdot 10^{4}$ atoms·g$^{-1}$. The average annual $^{10}$Be/$^{7}$Be ratio for XA was $2.22 \pm 0.12$, and
the average for the ZQL site was $1.62 \pm 0.08$ (Fig. 1c). The complete dataset is given
in Tables S3 and S4.
The XA sampling site is a typical high-dust locality on the Loess Plateau, and the
QL sampling site is relatively humid with lower dust content. Both sites experience
distinct similar seasonal changes. $^{7}$Be and $^{10}$Be aerosol concentrations show large
fluctuations associated with precipitation (Fig. 1a, 2b). The $^{10}$Be concentrations and
$^{10}$Be/$^{7}$Be ratio from dry deposition at the XA site are significantly higher than those at
the QL site, as well as coastal areas near the same latitude during the same period





(such as Dazaifu and Tokyo in Japan (Yamagata et al., 2019)), especially in the dry
season of the Loess Plateau, such as spring. QL serves as a control site with relatively
higher precipitation and is located in the Qinling National Nature Reserve with a high
normalized vegetation index (NDVI) (He et al., 2019). QL is only 60 km away from
XA. As shown in Fig. 1c, the $^{10}$Be/$^{7}$Be ratio of dry and wet deposition at the two
sampling sites during wet periods is consistent and reflects the amount of deposition.
Aerosol $^{10}$Be/$^{7}$Be ratios at the two sites during dry periods are higher than the
rainwater $^{10}$Be/$^{7}$Be from the same period. This phenomenon is more obvious at dusty
XA than at QL. Fig. 1d shows that the mean TSP concentration at XA (234.20 μg·m$^{-3}$)
was higher than the mean at QL (119.94 μg·m$^{-3}$). The average aerosol Al content at
XA (3.6 μg·m$^{-3}$) is significantly higher than at QL (1.5 μg·m$^{-3}$) and in coastal areas
(the mean in Japan is approximately 2.31 μg·m$^{-3}$ (Yamagata et al., 2010)), which
confirms that dust from the XA site has a higher probability of re-suspension.
XA is located in a warm temperate area with a semi-humid and semi-arid
continental monsoon climate. Its rainy seasons are unevenly distributed, and the
annual average precipitation is 500-750 mm (Chen et al., 2020b). The East Asian
winter monsoon and high-altitude westerly jets that carry dust from the western and
northern deserts profoundly impact the supply of aeolian materials on the Loess
Plateau (Shen et al., 2010). Dust-borne $^{10}$Be from resuspended soil dust contributes
significantly to the surface aerosol $^{10}$Be, and obscures atmospheric mass movement
information such as the STE (Fig. 1a-c). As expressed in equation (1), to obtain the
actual deposition flux and associated $^{10}$Be and $^{10}$Be/$^{7}$Be, it is necessary to remove
dust-borne interference, as discussed below.

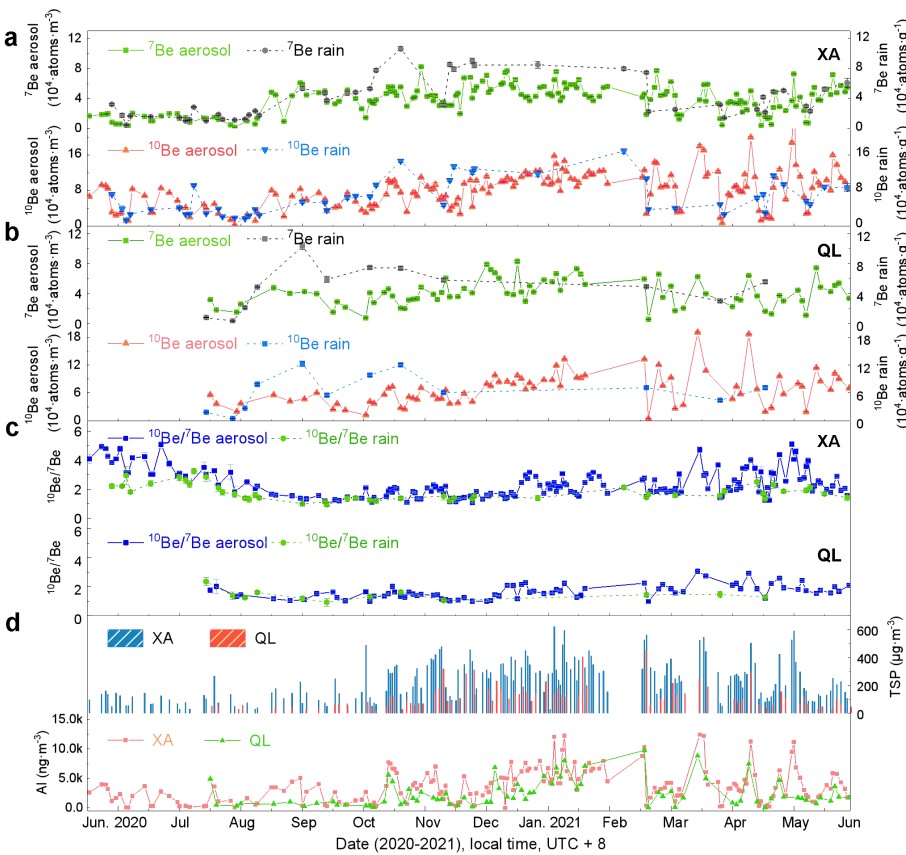

**Fig. 1** $^7$Be and $^{10}$Be concentrations and $^{10}$Be/$^7$Be ratios in the atmosphere above the
Loess Plateau. Daily variation in $^7$Be and $^{10}$Be concentrations at XA (a) and at QL (b).
(c). Daily $^{10}$Be/$^7$Be ratios (wet and dry deposition) of the Loess Plateau. (d). Daily
atmospheric TSP content and aerosol Al content at XA and QL.

**3.2 Correction for dust-borne $^{10}$Be on aerosol $^{10}$Be/$^7$Be values**

Due to its long half-life (1.36×10$^6$ years), $^{10}$Be accumulates after being deposited

on the ground, and $^{10}$Be abundances per gram of soil are approximately 10$^4$ times





higher than corresponding $^{10}$Be abundances in the near-ground atmosphere. In
contrast, $^{7}$Be can not accumulate in the soil due to its relatively short half-life (53.3
days) (Yamagata et al., 2010). Therefore, resuspended dust-borne $^{10}$Be can increase
the atmospheric $^{10}$Be concentration and $^{10}$Be/$^{7}$Be ratios. Dust-borne $^{10}$Be can be
corrected by considering all of the sources of $^{10}$Be in the bulk measurements:

$$[^{10}Be]_{bulk} = [^{10}Be]_{bg} + [^{10}Be]_{STE} + [^{10}Be]_{dust} \quad (2)$$

$$[^{10}Be]_{corr} = [^{10}Be]_{bg} + [^{10}Be]_{STE} \quad (3)$$

The measured aerosol $^{10}$Be concentration, $[^{10}Be]_{bulk}$ contains three distinct $^{10}$Be
components (equation 2). The background value $[^{10}Be]_{bg}$ refers to the $^{10}$Be
concentration in the surface-atmosphere at equilibrium with no STE component.
$[^{10}Be]_{STE}$ is the $^{10}$Be carried by the STE, and $[^{10}Be]_{dust}$ is resuspended dust-borne $^{10}$Be
(all concentrations in atoms·m$^{-3}$). The dust-borne $^{10}$Be must be removed to calculate
the corrected value $[^{10}Be]_{corr}$ (equation 3).
We measured atmospheric $^{7}$Be and $^{10}$Be values from different sites (Fig. 2a) and
different seasons (Fig. 2b), and used equation (1) to eliminate the influence of soil
dust on atmospheric $^{10}$Be (Fig. 2c). The resuspension of terrestrial dust is controlled
by dryness and wind power (Zhang et al., 1997). Therefore, we chose representative
sampling sites with different atmospheric relative humidity and NDVI characteristics.
These sampling sites include ZW (h) in an arid and high-dust area, XA (b) and TY (i)
in semi-arid and low-humidity dusty regions, and CD (j), NN (k), LYG (l), and QL (g)
in humid and low-dust areas. These areas include typical geographic environments
such as deserts, coastal regions, humid inland regions, and areas prone to drought.


The $^{7}$Be concentration in the atmosphere is minimally disturbed by dust
(Yamagata et al., 2010). However, as shown in Fig. 2b, in three different seasons
(autumn, winter, and spring), $^{10}$Be concentration fluctuations caused by the influence
of soil dust in different regions in the same season or at different times in the same
region are relatively large, which leads to large fluctuations in observed atmospheric
$^{10}$Be/$^{7}$Be ratios (darker columns in Fig. 2c). These results show that the drier the area
is, the greater the short-term fluctuation of $^{10}$Be/$^{7}$Be (Fig. 2c). Furthermore, Chen et al.
(2020b) pointed out that soil $^{10}$Be in areas with less precipitation ($< 1200$ mm·a$^{-1}$) is
more likely to be resuspended and not deposited in the surface soil. This conclusion
also shows once again that different amounts of resuspended dust can cause
fluctuations in observed atmospheric $^{10}$Be concentrations.
Based on equation (1), after removing the soil dust contribution of each sampling
site (contamination rate shown as the black line, the bottom panel of Fig. 2c), the
corrected average $^{10}$Be/$^{7}$Be ratios (colored horizontal lines, upper panel of Fig. 2c)
from each region are very similar. Among them, the XA $^{10}$Be/$^{7}$Be correction ratio (b)
in winter (January) is larger than the average value, caused by local STE events. The
$^{10}$Be/$^{7}$Be correction is relatively large, and the relative average value fluctuates
significantly in spring (April), because STE events frequently occur in spring. The
autumn-winter-spring trend in the average $^{10}$Be/$^{7}$Be correction value is also consistent
with the seasonal variation of the $^{10}$Be/$^{7}$Be ratio. Detailed $^{7}$Be and $^{10}$Be results are
given in Table S5.
To correct for dust-borne $^{10}$Be concentrations in the atmosphere, the soil erosion



293 conditions for each region need to be considered. Early studies made this correction

294 based on Ca/Mg content (Brown et al., 1989;Mann et al., 2011) or U isotope

295 composition (Monaghan et al., 1986). However, these methods overestimate the

296 effects of dust and rely on assumptions that pose weak constraints on the dust

297 composition (Graham et al., 2003). Zhang et al. (1994) pointed out that the Al

298 provides an excellent means to calculate soil dust flux. Through simultaneous

299 observations in different areas, we confirmed that the Al flux method (equation 1)

300 proceeds an effective correction for atmospheric $^{10}$Be. The corrected atmospheric $^7$Be

301 and $^{10}$Be yielded $^{10}$Be/$^7$Be records that reflected daily subsidence levels as an indicator

302 of surface deposition processes. At the same time, the corrected atmospheric $^{10}$Be

303 provides a practical observation method for quantifying the $^{10}$Be atmospheric input

304 (Yi et al., 2019a), and to study East Asian summer monsoon rainfall changes and

305 geomagnetic field changes (Kong et al., 2020).

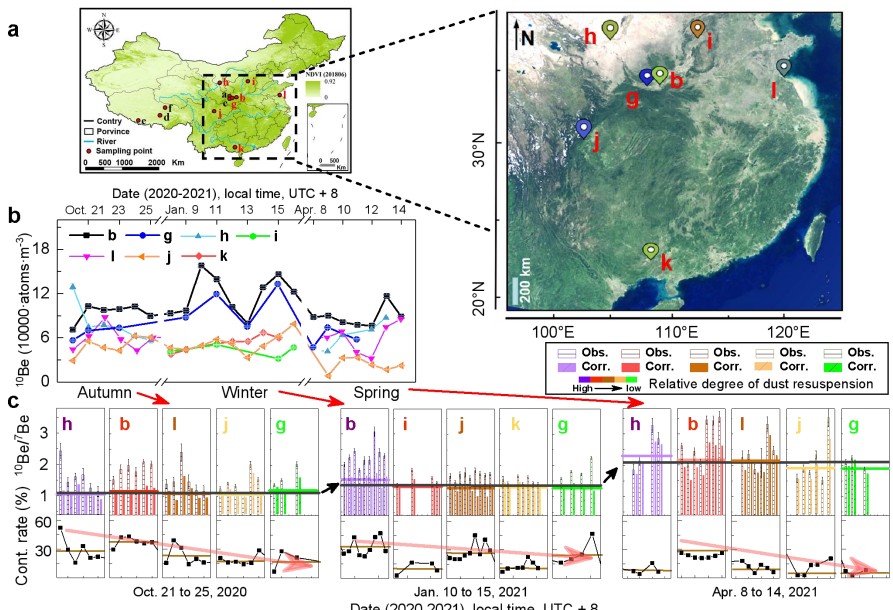

**Fig. 2** Atmospheric $^{10}$Be/$^7$Be observations and the impact of dust-borne $^{10}$Be from each study region. (a). Sample site map with NDVI (modis.gsfc.nasa.gov/), and a satellite image of the entire study area (from © Google Maps). (b). Short-term fluctuations in surface aerosol $^{10}$Be concentration during 3 different seasons at each sampling site. (c). Observed aerosol $^{10}$Be/$^7$Be values (darker hollow columns), corrected $^{10}$Be/$^7$Be values (brighter solid columns), and dust contribution (lower panel) before and after correction in 3 different seasons and regions. The horizontal lines represent average $^{10}$Be/$^7$Be values after correction and the average value contribution rate from dust-borne $^{10}$Be. The color of the columns from dark to light indicates that the relative resuspension amount of dust in various places in the same season gradually decreases. The arrows indicate trends.

**3.3 Contribution of soil dust $^{10}$Be in aerosols on the Loess Plateau**

The observed aerosol $^{10}$Be/$^7$Be ratios at the XA site during the dry period (blue



line in Fig. 3a) deviate significantly from recent rainwater $^{10}$Be/$^{7}$Be ratios (green line
in Fig. 3a). In contrast, aerosol $^{10}$Be/$^{7}$Be observations from the relatively low-dust QL
site are consistent with rainwater $^{10}$Be/$^{7}$Be (Fig. 3a). This result highlights the
importance of correcting for resuspended dust-borne $^{10}$Be.

The corrected aerosol $^{10}$Be/$^{7}$Be ratios (red line in Fig. 3a) follow the rainfall

$^{10}$Be/$^{7}$Be trend. Corrected aerosol $^{10}$Be/$^{7}$Be values from XA and QL are 0.91-3.73 and
0.93-2.56, respectively. The average annual contributions of resuspended dust-borne
$^{10}$Be for XA and QL were ~24% and ~12%, respectively (Fig. 3b; Table S3 and S4).
The contribution of soil dust from the XA site is much higher than in low dust areas,
such as New Zealand (11%) (Graham et al., 2003), Japan (~15%) (Yamagata et al.,
2010), or Seville (10%) (Padilla et al., 2019).

The corrected aerosol $^{10}$Be/$^{7}$Be ratios (red line in Fig. 3a) remove abrupt transient

changes associated with dust-borne $^{10}$Be (not STE events. This fluctuation is
especially apparent at the XA site and is seen as "V"-shaped changes in the
uncorrected record (enlarged view of Fig. 3a). The corrected $^{10}$Be/$^{7}$Be ratios seen at
the XA and QL sites are very similar, as would be expected once the dust-borne $^{10}$Be
has been removed. In addition, for atmospheric motion information that cannot be
captured by rain samples when there is no precipitation, these corrected dry
deposition $^{10}$Be/$^{7}$Be ratios are an effective supplement.

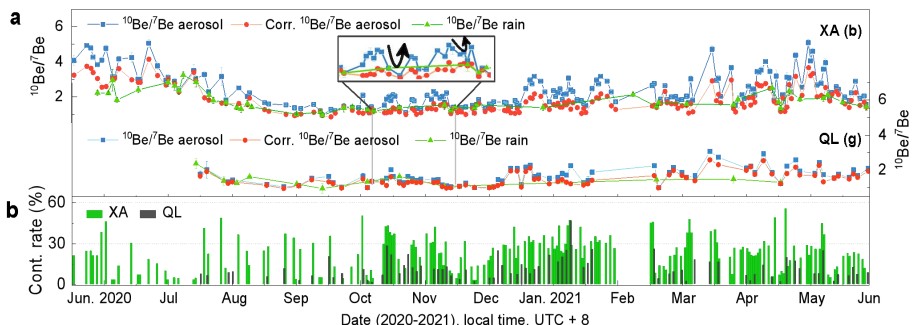

**Fig. 3** Correction of resuspended soil dust in aerosol $^{10}Be/^{7}Be$ on the Loess Plateau.

(a). Comparison of aerosol $^{10}Be/^{7}Be$ observations (blue) and corrected values (red),

with rainwater $^{10}Be/^{7}Be$ (green). (b). The contribution of soil dust $^{10}Be$ to aerosol

$^{10}Be/^{7}Be$ at the XA site (green) and QL site (black).

**3.4 Stratospheric air intrusions on the Loess Plateau and their contribution to surface O₃**

As shown in Fig. 1a and 1b, high $^{7}Be$ and $^{10}Be$ concentrations in winter are

caused by the relatively long-term accumulation of surface aerosols during periods of

low precipitation (Yamagata et al., 2019). However, $^{10}Be/^{7}Be$ ratio change relatively

little in these cases (Fig. 1c). This is because the fixed production rate ratio and the

similar geochemical properties make $^{10}Be/^{7}Be$ resistant to air mass dilution at

different latitudes and rain removal. In addition, the large difference in the half-lives

of $^{7}Be$ and $^{10}Be$ and their fixed production rate lead to stratospheric $^{10}Be/^{7}Be$ ratios

that are significantly larger than in the troposphere. Thus, $^{10}Be/^{7}Be$ increases seen near

the surface air can be used to detect stratospheric air intrusion events.

Fig. 4a shows a distinct seasonal pattern in $^{10}Be/^{7}Be$ ratios from the XA and QL





sites. This record is consistent with observational records (Jordan et al., 2003). Higher
$^{10}Be/^{7}Be$ ratios can indicate stratospheric air intrusion events superimposed on an
annual mean value of ~1 (Fig. 4). In spring and early summer (March to July), the
$^{10}Be/^{7}Be$ values reach a maximum of approximately 3 times higher than the annual
average. The increase is mainly affected by the Brewer-Dobson circulation in the
northern hemisphere (Butchart, 2014). The monthly $^{10}Be/^{7}Be$ variation box plot in the
upper right corner of Fig. 4a shows that the $^{10}Be/^{7}Be$ ratio between March and July is
relatively large and that the high-frequency positions are scattered. This indicates that
the intensity and frequency of stratospheric air intrusions during the "spring leakage"
of the atmosphere are higher. Conversely, the frequency and intensity of $^{10}Be/^{7}Be$
changes observed in other months are relatively scattered, indicating fewer but
distinct stratospheric air intrusion events during other periods of the year.

Sunlight is a natural prerequisite for the photochemical generation of $O_3$ at the

earth's surface (Kondratyev and Varotsos, 1996). The sunlight radiation follows a
symmetrical intensity law in the spring and autumn at the XA site (Fig. 4c). However,
surface $O_3$ concentrations at the XA site (Fig. 4b) are inconsistent with sunlight
radiation intensity (Fig. 4b blue area). This feature is consistent with the seasonal
increase in $^{10}Be/^{7}Be$. We suggest that surface $O_3$ concentration increases reflect an
influx of stratospheric $O_3$, indicated by frequent and relatively large changes in the
$^{10}Be/^{7}Be$. Similar results have been confirmed during spring in other regions using
ground-based lidar observations (Langford et al., 2009), airborne observations
(Weigel et al., 2012), and atmospheric models (Zhao et al., 2021). This also explains





why surface $O_3$ concentrations are elevated in spring when the light intensity is
comparable to that in autumn. With the arrival of summer, the light intensity gradually
increases, and $O_3$ becomes dominated by the photochemical process at the surface
(Fig. 4b).

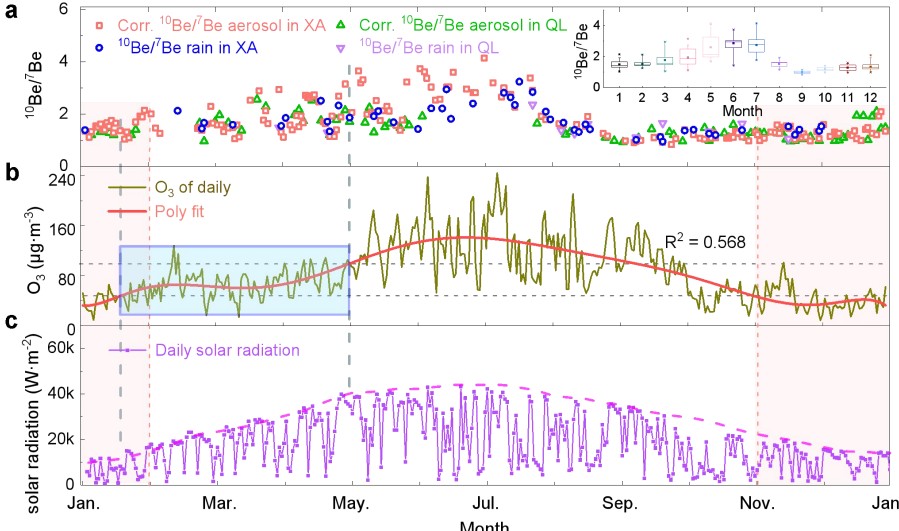


**Fig. 4** The $^{10}Be/^{7}Be$ ratio response to seasonal stratospheric air intrusions, consistent
with seasonal surface $O_3$ production on the Loess Plateau. (a). Seasonal $^{10}Be/^{7}Be$
variations. Inset shows a box plot (25%-75%) of monthly atmospheric $^{10}Be/^{7}Be$, with
the 5%-95% whiskers, and minimum/maximum values indicated by asterisks. (b).
Seasonal $O_3$ variations near the surface air at the XA site. Blue box indicates periods
of high surface $O_3$ concentrations that are higher than expected from corresponding
sunlight radiation intensities. (c). Seasonal sunlight radiation intensity at the XA site,
with associated trend line (red dashed line).



### 3.5 Evaluation of stratospheric $O_3$ intrusion with weak STE in winter


In addition to the strong "spring leakage" stratospheric air intrusion events that
occur regularly every year, a number of relatively low-intensity stratospheric air
intrusions are known to occur in other seasons. Li et al. (2021) pointed out that $O_3$
pollution during the winter haze season on the North China Plain is increasing. Ozone
pollution caused by winter stratospheric air intrusions is particularly serious (Chen et
al., 2020a). Although low-intensity and rapid stratospheric air intrusions are less
significant than in summer, their adverse effects cannot be ignored. Our study shows
that processes can be identified using ground-based $^{10}Be/^{7}Be$ measurements.
Traditional gamma spectrometry is less well-suited to obtaining high-precision $^{7}Be$
measurements of low volume air samples. Furthermore, our approach makes it
possible to make the daily measurements provide the requisite resolution necessary to
understand the rapid and transient chemical reactions in the atmosphere (Zheng et al.,

2011).

We observe frequent low but statistically significant ozone fluctuations in winter
on the Loess Plateau (pink shaded area, Fig. 4). A comparison with our $^{10}Be/^{7}Be$ data,
suggest low-intensity and rapid stratospheric air intrusions (November to January of
the following year) at the XA site (Fig. 5). In the absence of stratospheric air
intrusions, surface ozone shows a diurnal pattern with nighttime lows. However, when
$^{10}Be/^{7}Be$ increases (blue vertical shaded area in Fig. 5a), surface $O_3$ concentrations
increase (Fig. 5b). During these times, especially at night (shaded area in Fig. 5b),
surface $O_3$ concentrations are not observed to decrease, and diurnal ozone pattern





disappears. High $O_3$ concentrations are characteristics of the stratosphere (Kley et al.,
2007). Therefore, these phenomena may be related to stratospheric air intrusions.
Taken alone, the ozone trends can only be considered as indirect evidence for
stratospheric air intrusions. However, considered alongside the $^{10}Be/^7Be$ results at the
XA site, we interpret the covariation in winter to be related to rapid (1-3 days)
stratospheric air intrusions.

A fixed relationship between $O_3$ and $^7Be$ during stratospheric air intrusions was

proposed by Bazhanov and Rodhe (1997). Because $^{10}Be/^7Be$ avoids the interference
of tropospheric variability on $^7Be$ concentrations, we substitute $\Delta(^{10}Be/^7Be)$ for the
$\Delta^7Be$ to interpret surface ozone increases ($\Delta O_3$). The $\Delta O_3$ and $\Delta(^{10}Be/^7Be)$ values
show a significantly positive correlation ($R^2 = 0.889$, $p < 0.01$. Fig. S4), providing a
useful estimate of stratospheric ozone to the surface.

A surface $O_3$ record obtained by the above method is shown in Fig. 5c. We

estimate that surface $O_3$ increased by 25.45% in the XA site during the winter of
2020/21. Chen et al. (2020a) used the hemisphere WRF-CMAQ model to find that the
contribution of STE to surface $O_3$ in the winter of 2015 in eastern China was
approximately 15%-21%. Stratospheric air intrusion in the Qinghai-Tibet Plateau of
China has been shown to increase the $O_3$ concentrations in the troposphere by
approximately 53% (Zhang et al., 2021). The influence of weather on surface $O_3$ in
China varies by region, season, and year. It may be equivalent to or even more
significant than the impact of changes in anthropogenic emissions, of which
atmospheric motion has a significant contribution (Liu and Wang, 2020). $O_3$ brought



by stratospheric air intrusion seriously affects the chemical balance of the
tropospheric atmosphere. For example, increased night-time $O_3$ (mainly input from
the stratosphere under STE) accelerates the conversion of $NO_x$ to nitrate (Tang et al.,
2021), which has an important impact on atmospheric environmental pollution
process. In addition, this study proved that the relatively weak stratospheric air
intrusion process is randomly generated, which is an impact that needs to be
considered for the future surface $O_3$ concentration simulation.

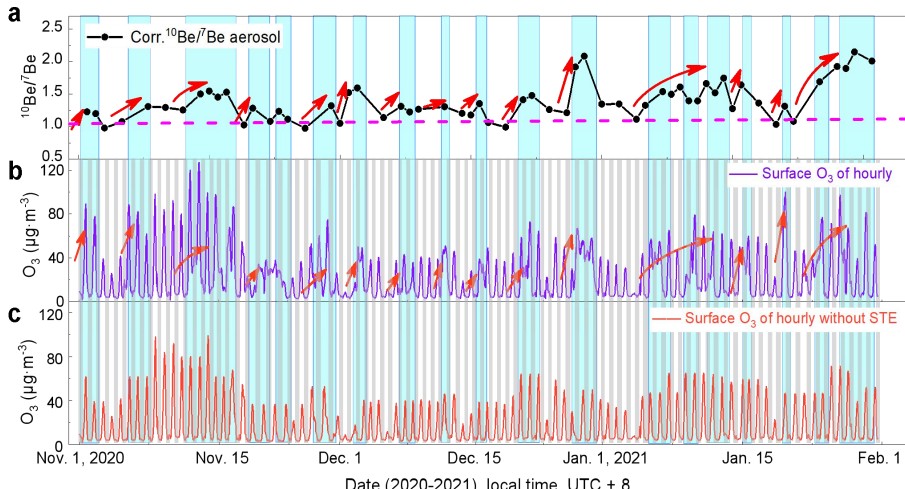


**Fig. 5** The influence of winter stratospheric air intrusions at the XA site and surface
$O_3$ compared with daily $^{10}Be/^7Be$ measurements. Aerosol $^{10}Be/^7Be$ ratio (a) and
surface $O_3$ concentrations (b) high-frequency variations at the XA site in winter. (c).
Surface $O_3$ concentrations after removing the contribution by stratospheric air
intrusions (see text). The purple line segment in Fig. 5a is the baseline of the $^{10}Be/^7Be$
ratio when no STE occurs. The blue shaded areas indicate stratospheric air intrusions



determined by $^{10}$Be/$^{7}$Be. Shaded areas in Fig. 5b and 5c indicate nighttime. Arrows
indicate trends.

**4 Conclusions**
STE is an important channel for transmitting atmospheric matters, particularly
low-intensity and frequent stratospheric air intrusion events that can quickly transmit
chemical matters to affect the environment. This work presents annual ground-based
$^{10}$Be/$^{7}$Be records documenting stratospheric air intrusions. The main features of these
records include the following:
• High-sensitivity measurements (error ~ 1.5%) of $^{7}$Be and $^{10}$Be from single
targets prepared from aerosols (down to ~200 m$^3$ air) depending on AMS.
• Results in different regions confirm that Al content can be used to correct
resuspended dust-borne $^{10}$Be in aerosol $^{10}$Be.
• By measuring both rainwater and aerosols, $^{10}$Be/$^{7}$Be measurements offer a
means of identifying stratospheric air intrusions throughout the year, with daily
resolution. The first of such records is presented herein, from the Loess Plateau of
China. We document $^{10}$Be/$^{7}$Be records for 2020-2021 from two sites: 1) XA site -
rainwater $^{10}$Be/$^{7}$Be was found to be 0.94-3.24, and the corrected aerosol $^{10}$Be/$^{7}$Be
range was found to be 0.91-3.73 after removal of the regional dust-borne component
(~24%); 2) QL site - rainwater $^{10}$Be/$^{7}$Be was found to be 0.94-2.36, and the corrected
aerosol $^{10}$Be/$^{7}$Be range was found to be 0.93-2.56 after the removal of the regional
dust-borne component (~12%).





• Atmospheric $^{10}$Be/$^{7}$Be on the Loess Plateau confirms that stratospheric air
intrusions occur frequently and rapidly (1-3 days) throughout the year, with the
strongest events in spring (March-July). These have a significant influence on surface
ozone. It is shown by $\Delta(^{10}$Be/$^{7}$Be) that even under weak stratospheric air intrusions in
winter, the cumulative contribution to surface ozone at the XA site in 2020/21 is
~25%.
**Data availability**
The author declares that the main data supporting the results of this study can be
found in the text and its supplementary materials.
**Supplement**
Supplementary information is available for this paper.
**Author contribution statement**
Xuke Liu finished the experiments, wrote the first version manuscript and
analysed the data. Yunchong Fu conceived the original idea, performed AMS analysis
and revised the initial manuscript. Xuke Liu and Yunchong Fu designed research
roadmap. Li Zhang, Yanting Bi, and Guoqing Zhao designed chemical experiments
and collected samples. Yunchong Fu assisted in AMS method development and
supervised the research project. All the authors discussed the results and commented
on the manuscript, and George S. Burr for revising manuscript language.
**Competing interests**
The authors declare no competing interests.



**Acknowledgements**
We thank Weijian Zhou, Feng Xian, Peng Cheng, LuYuan Zhang, and Qi Liu for
their help in the conducting experiments and discussing the results.
**Financial support**
This research has been supported by the National Natural Science Foundation of
China (Grant No. 11975240), the Strategic Priority Research Program of Chinese
Academy of Sciences (B) (Grant No. XDB40000000), and the Youth Innovation
Promotion Association CAS (Grant No. Y2021108).

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
