# Peer review of "Detection of Stratospheric Air Intrusion Events From Ground-based"

_Atmospheric Chemistry and Physics, 2022_

## Author Comment (AC1)

Note that reviewers' questions are in **black**, our response is in **blue**, and the changes made is in **red.**
* * *
Ref: No.: acp-2022-282

Title: Detection of Stratospheric Air Intrusion Events From Ground-based High-resolution $^{10}$Be/$^{7}$Be by Accelerator Mass Spectrometry

Journal: Atmospheric Chemistry and Physics

**We would like to thank reviewer #1 for the interest shown in our paper and the valuable feedbacks that will improve the quality of the paper. The responses to the comments of the reviewer one by one are as follows:**

**Reviewer #1:** Review of "Detection of stratospheric air intrusion events from ground-based high-resolution 10Be/7Be by accelerator mass spectrometry" by X K Liu et al. In this paper, the authors derive 10Be and 7Be measurements from collected aerosol and rainwater samples from 7 locations in central China. A large amount of data is presented. I will return to the question of the data coverage in my comments. The results are interesting and should be eventually published in final form, however, some details seems to be glossed over and need to be addressed. In my view, the paper needs major revision.

**Q1:** Abstract, line 30-32. Here, it says that 24% of 10Be in Xi'an (XA) air was dust-borne, yet the various figures 2 and 3 show considerable variability.

**A1:** We thank the reviewer for the question. Indeed, as the reviewer sees, the dust interference is fluctuating, and the ~24% here refers to the average value. For clarity, it has been modified as follows:

We show that for the city of Xi'an, the average proportion of dust-borne (resuspended) $^{10}$Be was about 24% in 2020/21.

**Q2:** line 48 delete "hard-to-capture". I am not sure what is meant here, but I assume you mean that it is difficult to quantify the intrusion events. Perhaps rewrite the sentence.

**A2:** We thank the reviewer for the suggestion. "hard-to-capture" has been removed.

**Q3:** line 57 "intrusions"

**A3:** We thank the reviewer for the question, it has been modified:

Traditional observation methods of stratospheric air intrusions mainly rely on....

**Q4:** line 64 replace "it is generally believed.."with "it has been suggested.."

**A4:** We thank the reviewer for the question, it has been modified:

...., and it has been suggested that ground-based $^7$Be analyses offer a means of identifying and tracing stratospheric air intrusions.

**Q5:** line 74 delete "the" before "very different"

**A5:** We thank the reviewer for the question, it has been modified.

**Q6:** Line 82-87. Here, it is stated that 7Be was done by AMS. This is not correct. **Some of the measurements were done by AMS**. In the supplementary data and lines 166-169, it is stated that "some .. membranes were selected and ¼ of them were measured by AMS. The 7Be in the remaining ¾ filter membranes…were analyzed by high-purity germanium (HpGe) detector(s)..and analyzed a 477.6KeV..". Maybe this was only a fraction of the samples, but please rewrite and explain what was actually done.

**A6:** We thank the reviewer for the question. The gamma spectrometry measurement is used here only to correct and check the 7Be-AMS measurement results, not to obtain the 7Be observation record. All the observational records of 7Be and 10Be presented in this work are obtained based on AMS measurements. For clarity, the following description has been added on the first sentence of the last paragraph of section 2.3:

In addition to $^7$Be being measured by AMS, we also correct and cross-checked the results of 7Be-AMS by gamma spectroscopy.

**Q7:** Line 105-106. It is stated here that "an annual record with daily resolution" was obtained. This is most true for site XA (Xi'an) and QL (Qinling) but even there, it's not actually daily, there are missing values.

**A7:** We thank the reviewer for the comment. The daily resolution here means that the sample sampling period is one day, some of which are continuous every day, and another part of the time is not continuous due to the sampling process such as impact of COVID-19.

**Q8:** Section 2.1 Samples from the other sites were only measured at a few times in October 2020, Jan 2021 and April 2021.

**A8:** We thank the reviewer for the question.The resuspension of terrestrial dust is controlled by aridity and wind power. Therefore, we chose representative sampling sites with different atmospheric relative humidity and NDVI characteristic to verify the general applicability of this calibration method. The distribution of sample collection times at these sampling sites is noted at the appropriate place in the manuscript:

Zhongwei (ZW, 37°30′57″N, 105°11′8″E), Taiyuan (TY, 37°48′26″N, 112°34′55″E), Chengdu (CD, 30°56′44″N, 103°40′41″E), Nanning (NN, 22°50′18″N, 108°16′51″E), and Lianyungang (LYG, 34°46′4″N, 119°26′55″E) for short-term intermittent monitoring (in October 2020, Jan 2021 and April 2021).

**Q9:** Section 2.3 lines 148-151 and supplement. 1-2. Here, it is stated that the "transmission after the second stripping foil" was improved to ~24%. If this is into 10Be4+, this is a remarkable improvement. Most labs get lower yields even into 3+. Also, what is the location of the second stripping foil? I assume it is after the analyzing magnet and before an ESA, but no details of the AMS setup are given. The discussion in the reference Zhang and Fu (2017) is not completely helpful. First, in that paper, the yield from Be2+ to Be3+ is given as 31% **and from Be3+ to Be4+ is 3%. If some other improvement was done, it is not clear from this paper.** Also, much of the paper of Zhang and Fu (2017) discusses other methods, such as using BeF3- as an injection beam. Hence, we still don't know from the paper presented here by Liu et al. which method was used.

**A9:** We thank the reviewer for the question. In this work, the $BeO^-$ to $Be^{2+}$ to $Be^{4+}$ method is used to perform $^{10}Be$-$^7Be$-AMS measurements (Zhang and Fu, 2017). Secondary stripping film (500 nm $Si_3N_4$) behind the 115° high energy analysis magnet. In most instances, the efficiency of $Be^{2+}$ to $Be^{3+}$ is ~31% ($^{10}Be$ standard sample) (Zhang and Fu, 2017).

The low efficiency of $Be^{2+}$ to $Be^{4+}$ is due to the establishment of our preliminary method in 2017, and the components behind the stripping film have not been carefully optimization. In particular, the role of the quadrupole lens behind the 65° ESA is underestimated. This work was carried out after the 7Be-AMS transmission efficiency

Note that reviewers' questions are in **black**, our response is in **blue**, and the changes made is in **red.**

was re-optimized in 2018. That is to say, the most important thing is to adjust that quadrupole lens (although the transmission efficiency is lower in the early work of 2017, AMS is a relative measurement, which ultimately only affects the statistical error of the final results). Based on this, in the $^{10}$Be analysis, the terminal voltage works at 2.5 MV, the efficiency of $^{10}$Be$^{2+}$ to $^{10}$Be$^{3+}$ is ~31%, and the efficiency of $^{10}$Be$^{2+}$ to $^{10}$Be$^{4+}$ is ~24%. An example from the same standard samples in this work, AMS measurements of $^{10}$Be$^{3+}$ and $^{10}$Be$^{4+}$ in the $^{10}$Be standard sample ($^{10}$Be/$^{9}$Be = 2.709·10$^{-11}$) are given below:

Table 1 AMS measurement raw results of $^{10}$Be$^{3+}$ and $^{10}$Be$^{4+}$ in $^{10}$Be standard samples of this work ($^{10}$Be/$^{9}$Be = 2.709×10$^{-11}$)

| No. | Measurement No. | Raw average of $^{10}$Be$^{3+}$ | Efficiency$_1$ | Raw average of $^{10}$Be$^{4+}$ | Efficiency$_2$ |
|---|---|---|---|---|---|
| 1 | Be2020-0313 | $8.51×10^{-12} ± 4.25×10^{-14}$ | 31.41% | $6.46×10^{-12} ± 3.20×10^{-14}$ | 23.85% |
|  | Be2020-0314 | $8.51×10^{-12} ± 5.20×10^{-14}$ | 31.41% | $6.54×10^{-12} ± 3.48×10^{-14}$ | 24.14% |
| 2 | Be2021-737 | $8.53×10^{-12} ± 6.63×10^{-14}$ | 31.78% | $6.70×10^{-12} ± 3.91×10^{-14}$ | 24.73% |
|  | Be2021-738 | $8.50×10^{-12} ± 5.98×10^{-14}$ | 30.79% | $6.74×10^{-12} ± 3.64×10^{-14}$ | 24.88% |
|  |  | **Mean** | **31.35%** | **Mean** | **24.40%** |

Corresponding instructions are added in Supplementary Information:

**Analysis of $^{7}$Be and $^{10}$Be by AMS**

First, both were extracted as BeO$^{-}$ particles from the ion source. Next, with the accelerator terminal voltage set at 2.5 MV, beryllium ions are accelerated and the +2 charge state is selected using a high-energy analysis magnet. Stripping efficiency of BeO$^{-}$→Be$^{2+}$ is approximately 47%. After the stable and rare nuclides were separated by the analysis magnet, the rare nuclides passed through a secondary stripping film (500 nm, Si$_3$N$_4$) are stripped to +4 charge state. The beryllium atoms are completely stripped, that is, BeO$^{-}$→Be$^{2+}$→Be$^{4+}$. Based on the Be$^{2+}$ to Be$^{4+}$ method we initially established in 2017 (Zhang and Fu, 2017), the stripping efficiency has been improved. The increase in efficiency is based on the carefully adjustments to the quadrupole lens behind the 65 ° ESA. Based on this, in the $^{10}$Be analysis, the terminal voltage works at 2.5 MV, the efficiency of $^{10}$Be$^{2+}$ to $^{10}$Be$^{3+}$ is ~31%, and the efficiency of $^{10}$Be$^{2+}$ to $^{10}$Be$^{4+}$ is ~24%.......

**Q10:** Line 182-188. Here, the values from previous work on 10Be have errors, it would be good to quote them here.

**A10:** We thank the reviewer for the suggestion. Here, we choose the mean value within the range of reported values in papers in this area as the topsoil 10Be concentration and cite their results. In addition, the choice the topsoil 10Be concentration value had little effect on the correction results, which mainly depends on the resuspension degree of soil dust (this is mainly calculated by the content of Al in the aerosol).

**Q11:** Line 234. "material"

**A11:** We thank the reviewer for the suggestion, it has been modified:

....and northern deserts profoundly impact the supply of aeolian material on the Loess Plateau.

**Q12:** Line 236. I think "atmospheric movement information" isn't a good term, please use something like "atmospheric circulation"

**A12:** We thank the reviewer for the suggestion. Modify it to atmospheric circulation: obscures degree of $^{10}$Be change brought by atmospheric circulation such as STE (Fig. 1a-c).

**Q13:** Line 247. I think the accepted value of the half-life is 1.38Myr.

**A13:** We thank the reviewer for the suggestion, its half-life has been revised to 1.39 (1.38) Myr:

Due to its long half-life ($1.39 \times 10^6$ years)......

**Q14:** Line 250. "cannot"

**A14:** We thank the reviewer for the suggestion, it has been modified:

$^7$Be cannot accumulate in the soil due to its relatively short half-life (53.3 days)

**Q15:** Line 265 "aridity" not "dryness"

**A15:** We thank the reviewer for the suggestion, it has been modified:

......by aridity and wind power.

**Q16:** Figures 2,3 and 4 have small font sizes and they can be hard to read. I suggest scaling up the figures in the revised version.

**A16:** We thank the reviewer for the suggestion. The font sizes for Figures 2, 3 and 4 have been enlarged.

**Q17:** Line 333-334. What are the "V" events and don't you have meteorological data about what happened on those days at XA and QL?

**A17:** We thank the reviewer for the question. The "V" event mentioned here refers to the change in the observed aerosol 10Be/7Be ratio in the days before, during and after the rain (enlarged view of Fig. 3a). That is, the 10Be/7Be ratio was high in the days before the rain, it decreased significantly during the rain, and the value began to rise again in the days after the rain stopped.This also shows that the interference of resuspended dust is relatively large in the loess area of China, and correction is necessary. We have supplemented this as follows:

This fluctuation is especially apparent at the XA site and is seen as "V"-shaped changes in the uncorrected record (enlarged view of Fig. 3a, which refers to the uncorrected aerosol $^{10}$Be/$^{7}$Be ratios change in the days before, during, and after the rain).

**Q18:** Line 402-406 restates what has already been said and can be deleted.

**A18:** Thanks to the reviewer for the suggestion, which has been removed.

**Q19:** Line 369. This seems to be a new subject here. It appears the authors try to argue that the ozone in the ground-based measurements at XA are not due to photochemical effects (e.g. due to pollution) and the ozone is due to stratospheric injection. This seems like a long shot, especially in a city like XA with considerable anthropogenic air quality problems, although I agree others (e.g. Langford) have assert similar effects elsewhere.

**A19:** We thank the reviewer for the question. We do not deny the contribution of photochemical effects to surface ozone here, but want to express that the contribution of stratospheric ozone input in this period is more obvious than that in other seasons.

In this work, for the seasonal intrusion process confirmed by other means, we reveal that the intrusion of stratospheric ozone in spring is significantly higher than in other seasons by the phenomenon that the 10Be/7Be ratio in spring is significantly higher

than that in other seasons (~3 times). We are sorry that our statement misled the reviewer's understanding. To give a clearer expression, modify the description in this paragraph as follows:

Sunlight is a natural prerequisite for the photochemical generation of $O_3$ at the earth's surface (Kondratyev and Varotsos, 1996). The sunlight radiation follows a symmetrical intensity law in the spring and autumn at the XA site (Fig. 4c). However, surface $O_3$ concentrations at the XA site (Fig. 4b) are inconsistent with regularity of sunlight radiation intensity (mainly in the spring represented in Fig. 4b blue area). This feature is consistent with the seasonal increase in $^{10}Be/^{7}Be$. We suggest that the abnormally increased part of the surface $O_3$ concentration may largely reflect the influx of stratospheric $O_3$, indicated by frequent and relatively large changes in the $^{10}Be/^{7}Be$. Similar seasonal intrusion results have been confirmed during spring in other regions using ground-based lidar observations (Langford et al., 2009), airborne observations (Weigel et al., 2012), and atmospheric models (Zhao et al., 2021).

**Q20:** Figure 4b. How are the data in figure 4b compared to fig. 4a? Are all the data used? 4b refers only to XA, but 4a also includes QL data.

**A20:** We thank the reviewer for the comment. Figure 4b refers to XA data only. The straight-line distance between XA and QL is about 60 km (described earlier in Section 3.1). The focus here is on stratospheric mass intrusion processes with large scales. In addition, there is no significant difference in the seasonal variation scale of surface ozone concentration and sunlight intensity between the two site. Therefore, only a weather station was set up in XA to collect solar radiation intensity data, and the ozone data at XA was downloaded from the China Meteorological Network. Added clarification in the legend to Figure 4 as follows:

**Fig. 4** ......Note: XA and QL are ~ 60 km apart, so the seasonal changes of sunlight radiation and ozone are represented by the XA site data.

**Q21:** Line 455. "STE is an important channel…". This sentence needs rewriting and makes little sense at present. Perhaps STE are important processes that can quickly transmit chemical material from the stratosphere into the lower atmospheric environment.."

**A21:** Thanks to the reviewer for the comment and suggestion. Modify the content as

suggested:

STE are important processes that can quickly transmit chemical material from the stratosphere into the lower atmospheric environment, particularly low-intensity and frequent stratospheric air intrusion events that can transmit chemical matters more quickly to affect the environment.

suggested:

STE are important processes that can quickly transmit chemical material from the stratosphere into the lower atmospheric environment, particularly low-intensity and frequent stratospheric air intrusion events that can transmit chemical matters more quickly to affect the environment.

---

## Author Comment (AC3)

Ref: No.: acp-2022-282

Title: Detection of Stratospheric Air Intrusion Events From Ground-based High-resolution $^{10}$Be/$^{7}$Be by Accelerator Mass Spectrometry

Journal: Atmospheric Chemistry and Physics

**We thank Professor Guan for his comments on our paper, and it is an honor to receive your attention to our work. We reply to your questions one by one as follows:**

Q1: The description of how accelerator mass spectrometry measures 7Be is not detailed enough. 7Be does not have a standard sample, how to measure it, and how to calculate it, it is not clear.

A1: The measurement of 7Be-AMS mainly adopts the method we used in 2017 (detailed explanation in the Zhang and Fu 2017, doi: 10.1088/1674-1137/41/1/018201). Among them, the correction for 7Be is based on 10Be standard as an internal correction, and then the correction is combined with the gamma spectrum. 10Be/7Be ratio is directly obtained by AMS measurement. In addition, for the measurement process of 10Be-7Be-AMS, combined with the comments of reviewer #1, relevant instructions have been added to the supplementary materials:

**Analysis of $^{7}$Be and $^{10}$Be by AMS**

First, both were extracted as BeO$^-$ particles from the ion source. Next, with the accelerator terminal voltage set at 2.5 MV, beryllium ions are accelerated and the +2 charge state is selected using a high-energy analysis magnet. Stripping efficiency of BeO$^-$→Be$^{2+}$ is approximately 47%. After the stable and rare nuclides were separated by the analysis magnet, the rare nuclides passed through a secondary stripping film (500 nm, Si$_3$N$_4$) are stripped to +4 charge state. The beryllium atoms are completely stripped, that is, BeO$^-$→Be$^{2+}$→Be$^{4+}$. Based on the Be$^{2+}$ to Be$^{4+}$ method we initially established in 2017 (Zhang and Fu, 2017), the stripping efficiency has been improved. The increase in efficiency is based on the carefully adjustments to the quadrupole lens behind the 65 ° ESA. Based on this, in the $^{10}$Be analysis, the terminal voltage works at

2.5 MV, the efficiency of $^{10}Be^{2+}$ to $^{10}Be^{3+}$ is ~31%, and the efficiency of $^{10}Be^{2+}$ to $^{10}Be^{4+}$ is ~24%. Secondary stripping technology effectively eliminates ($^{7}Li$) or reduces ($^{10}B$) interference from the isobar. The $^{10}Be/^{9}Be$ of the sample is calibrated according to the standard ($^{10}Be/^{9}Be$ = 2.709·10$^{-11}$) in Nishiizumi et al. (2007) . The measured energy spectrum is shown in Fig. S3.

Q2: The work in the article involves two regions, two samples, one year of collection and isotopic beryllium analysis. This is not the case in actual work, not every day, and part of the time data is not available and needs to be clearly described.

A2: We thank the reviewer for the comment. The daily resolution here means that the sample sampling period is one day, some of which are continuous every day, and another part of the time is not continuous due to the sampling process such as impact of COVID-19. The continuous distribution of the data can also be seen from the data graph in Figure 1.

Q3: Are suspended particles in rainwater removed prior to analysis? Why the rain sample does not contain the re-suspending dust 0.5h hours after the initiated? Both STE exchange derived and tropospheric generated 10Be are associated to air particles, not as gas, where the resuspension dust is also particles, which might be also diffused to a few thousands meters height, is there any evident showing that re-suspension particles (dust) can be removed in the first half hour rain precipitation, but not the particles that TSE originated do? Although these issues are mentioned in the manuscript, a clear description will help improve the quality of the manuscript.

A3: Thanks to the reviewer for their attention to the issue of resuspension dust interference. For the rainwater sample collection, the initial rainwater samples is removed (described in 2.1 Sample collection), which can effectively avoid the interference of the resuspended dust on the rainwater sample(Graham et al., 2003). Here, a rainwater sample was used as a control for resuspended dust interference

correction. Due to the limitation of the random occurrence of the rainfall process, the research of the tracer STE is more based on aerosol samples. stratospheric 10Be/7Be ratios much larger than 10Be/7Be ratios observed near the surface air. When stratospheric air intrudes, the 10Be/7Be ratio in the atmosphere at the earth's surface increases. In addition, the "V" event shown in Figure 3, The "V" event mentioned here refers to the change in the observed aerosol 10Be/7Be ratio in the days before, during and after the rain. That is, the 10Be/7Be ratio was high in the days before the rain, it decreased significantly during the rain, and the value began to rise again in the days after the rain stopped.This also shows that the interference of resuspended dust is relatively large in the loess area of China, and correction is necessary. In addition to the comments of other reviewers, the description of the second paragraph of Section 3.3 of the manuscript has been revised as follows:

The corrected aerosol 10Be/7Be ratios (red line in Fig. 3a) remove abrupt transient changes associated with dust-borne 10Be (not STE events). This fluctuation is especially apparent at the XA site and is seen as "V"-shaped changes in the uncorrected record (enlarged view of Fig. 3a, which refers to the uncorrected aerosol 10Be/7Be ratios change in the days before, during, and after the rain).